# Deviation of Latitude Cut: A Simple Sign to Differentiate Total Hip Arthroplasty from Hemiarthroplasty in Radiography

**DOI:** 10.3390/jcm12196218

**Published:** 2023-09-26

**Authors:** Sunghoon Park, Jae Sung Yun, Dong-Ho Bang, Yongjun Jung, Kyu-Sung Kwack, Jung-Taek Kim

**Affiliations:** 1Department of Radiology, Ajou University School of Medicine, Ajou Medical Center, Suwon 16499, Republic of Korea; 2Musculoskeletal Imaging Laboratory, Ajou University Medical Center, Suwon 16499, Republic of Korea; 3Department of Radiology, Hankook General Hospital, Cheongju 63183, Republic of Korea; 4Department of Orthopaedic Surgery, Ajou University School of Medicine, Ajou Medical Center, Suwon 16499, Republic of Korea

**Keywords:** radiograph, hemiarthroplasty, total hip arthroplasty

## Abstract

This study aims to propose latitude cut deviation for differentiating hip arthroplasty types and evaluate its diagnostic utility in distinguishing total hip arthroplasty (THA) from hemiarthroplasty using radiography. After assessing various cup designs from top manufacturers for THA and hemiarthroplasty, we conducted a retrospective study on 40 patients (20 THA and 20 hemiarthroplasty). Three readers independently evaluated the radiographs, assessing acetabular sparing, cup–bone interface texture, and latitude cut deviation. Diagnostic performance and inter-observer agreement were compared using receiver operating characteristic curves and the Fleiss kappa coefficient. Latitude cut deviation measured on implant designs ranged from 19% to 42% in hemiarthroplasty and from −12% to 9% in THA. The sensitivity, specificity, and accuracy used to distinguish THA from hemiarthroplasty were 60–85%, 55–95%, and 62.5–77.5% for acetabular sparing; 100%, 50–80%, and 75–90% for cup–bone interface texture; and 100%, 90–100%, and 95–100% for latitude cut deviation. Inter-observer agreement for acetabular sparing, cup–bone interface texture, and latitude cut deviation ranged from moderate to excellent (κ = 0.499, 0.772, and 0.900, respectively). The latitude cut deviation exhibited excellent diagnostic performance and inter-reader agreement in distinguishing hemiarthroplasty from THA on radiographs, offering a concise way to identify hip arthroplasty type.

## 1. Introduction

Over the past 50 years, hip arthroplasty has become a common procedure, with hemiarthroplasty and total hip arthroplasty (THA) being the most frequently employed forms of hip arthroplasty. Furthermore, both implants and surgical methods have steadily improved, and different implants are used to treat various conditions in hip arthroplasty, ranging from resurfacing implants to mega-prostheses [1] (Figure 1). Hemiarthroplasty involves the replacement of the native femoral head and neck with an artificial prosthesis, leaving the native acetabulum untouched [2]. In THA, both the femoral and acetabular articular surfaces are replaced with artificial prostheses [3].

Owing to differences in structure, THA and hemiarthroplasty have characteristic complications. The erosion of the acetabular cartilage and bone is a common late complication of hemiarthroplasty, with a reported incidence of up to 66% [4]. Such erosion can lead to pain and component migration and may require revision [5]. On the other hand, aseptic loosening of the acetabular metal cup is a complication of THA and represents the most common cause of revision surgery, accounting for 75% of cases [6].

Radiography is a standard, basic examination for evaluating hip arthroplasty and identifying postoperative complications [3]. Radiography is not only simple, less expensive, and less affected by artifacts but also provides higher image resolution and takes less time than computed tomography or magnetic resonance imaging. The initial step in recognizing complications in radiography is the determination of the type of replacement [7]. Radiologists and orthopedic surgeons can easily distinguish the two types of prostheses based on patients’ previous surgical records or serial postoperative images. Nevertheless, the retrieval of surgical records or serial postoperative images at each time when post-arthroplasty images are encountered may lead to intricacy and complexity in the image interpretation process. Additionally, records may not be available for some cases in which surgery was performed at another institution, or they may have been discarded as outdated. This forces readers to solely rely on diagnostic images for differentiation. Hip prostheses consist of a spherical cup and femoral stem around the hip joint on radiographs because of the similarity in contours between the two types of hip prostheses, distinguishing one from the other can be difficult.

Previous studies have reported several radiographic features that could distinguish hemiarthroplasty from THA, including the texture of the cup–bone interface, cup size, presence of screws or screw holes in the cup, and preservation of the subchondral bone plate of the acetabulum and adjacent cartilage joint space [1,2,8]. However, some findings, such as the texture of the cup–bone interface and the presence of screws or screw holes, are not always detectable in images. Moreover, cup size is a slightly vague indicator. Assessment of the adjacent cartilage joint space can also lead to misinterpretation because it may be affected by complications of arthroplasty; in particular, relying solely on the adjacent cartilage joint space for differentiation may result in misinterpretation of hemiarthroplasty with the erosion of the acetabular cartilage as THA [8]. In contrast, aseptic loosening or osteolysis at the bone–prosthesis interface in THA may be mistaken for adjacent cartilage joint space, mimicking a normal cup in hemiarthroplasty [2]. None of the reports focused on the geometrical difference in spherical contour.

In contrast to previous studies that relied on extrinsic features or features that may not always exist, this current study focused on the inherent features of the prosthesis originating from the implant design rationale. The present investigation was a two-part study. First, we explored the various cup designs offered by top manufacturers for THA and hemiarthroplasty, with a focus on the outer head in hemiarthroplasty and the acetabular cup in THA. Second, based on the results of the first part of this study, we compared the diagnostic values of cup geometry and other features to distinguish hemiarthroplasty from THA on postoperative radiographs.

## 2. Materials and Methods

### 2.1. Latitude Cut Deviation

Typical post-arthroplasty radiographs show a metallic silhouette consisting of a combination of spherical and longitudinal structures. Both types of hip arthroplasty use the femoral stem, which creates a longitudinal silhouette; however, the difference comes from the remaining part of the arthroplasty, which creates a spherical contour.

In this present study, the latitude cut deviation was examined to quantify the difference in the spherical silhouette. A circle that best matched the outer contour of the sphere was drawn, and subsequently, a chord line connecting the two points where the silhouette left the circle was also drawn. The latitude cut deviation was defined as the position of the chord line relative to the equator of the sphere. For investigational purposes, the parameter was quantitatively assessed as the “percentage of latitude cut deviation”, which was defined as the ratio of two distances. The numerator was the distance between the center of the circle and the chord line, whereas the denominator was the radius of the circle drawn in the first step. Instead of measuring two respective distances, the angle made with the chord line and center of the circle (∠ABC) was measured using ImageJ software (1.53o, http://rsb.info.nih.gov/ij/ (accessed on 24 January 2022)) to minimize the chance of measurement error (Figure 2).
Percentage of latitude cutoff deviation = CD/CB = sin (∠ABC)(1)

(If the chord line was below the equator, the percentage was positive; otherwise, it was negative).

### 2.2. Measurements of Latitude Cut Deviation on Commercially Available Cups in Hemiarthroplasty and THA

Given that implants are packed in sterilized packages, direct measurement of all size variations was not possible. Measurements made with a 3-dimensional computer-aided design program on the blueprints of implants offer better accuracy; hence, the major manufacturers of hip arthroplasty implants were requested to measure the percentage of latitude cut deviation in their blueprints. Two companies responded, namely DePuy Synthes (Warsaw, IN, USA) and Corentec (Cheonan, Republic of Korea). For those that did not (Stryker, Mahwah, NJ, USA, and Zimmer Biomet, Warsaw, IN, USA), the percentage of latitude cut deviation was digitally measured on the acetate templates of implants. 

For digital measurement, the templates were scanned at 300 dpi, and the digitized images were imported into PowerPoint (Microsoft, Redmond, WA, USA). A circle with 50% transparency was superimposed on each cup size to determine the circle that best fitted the outer contour of the cup. The rest of the measurement was the same as that previously described. 

The results indicated that the mean percentage of latitude cut deviation was 28.0% ± 5.3% (range: 19% to 42%) in hemiarthroplasty and 1% ± 3.9% (range: −12% to 9%) in THA (Table 1). The results of the first part of the investigation clearly showed the difference in latitude cut deviation between the two types of cups. Based on these results, we hypothesized that clinicians could differentiate the latitude cut deviation between the two types of implants via visual assessment of the latitude cut deviation without precise percentage measurement.

### 2.3. Comparison of Diagnostic Performance of Three Discriminators

#### 2.3.1. Patients

This study was conducted in accordance with the principles embodied in the Declaration of Helsinki. The second part of this study was approved by the Institutional Review Board of our institution, which waived the requirement for the acquisition of informed consent from study participants (AJIRB-MED-MDB-21-546).

To test our hypothesis, we planned to construct an image set. A database search of all patients who visited an orthopedic surgeon for follow-up or consultation from other institutions or clinics after hip arthroplasty from January 2020 to December 2020 was performed. The inclusion criterion was the availability of standing anteroposterior hip radiographs. The exclusion criteria were (i) the types of hip arthroplasty not related to this investigation, such as resurfacing and metal augmentation for massive bone defects around the acetabulum; (ii) images with definitive signs, such as a markedly rough texture of the cup–bone interface, visible screw holes, and a screw in the spherical contour; and (iii) inadequate imaging studies due to suboptimal image quality (Figure 3). After applying the inclusion and exclusion criteria, a total of 226 patients were identified in the database.

Using images and surgical records, a musculoskeletal radiologist and an orthopedic surgeon categorized the images into four groups. Because images depicting complications are challenging to acquire, we initially selected images of hemiarthroplasty with acetabular erosion and THA with acetabular loosening and subsequently gathered 10 cases for each category. Appendix A presents a breakdown of implants used in images.

#### 2.3.2. Radiograph Examination

Standing anteroposterior hip radiographs were obtained according to a standard radiographic protocol as part of routine clinical follow-up. On optimally positioned anteroposterior hip radiographs with neutral rotation, the coccyx should be directly in line with the pubic symphysis, whereas the iliac wings, obturator foramina, and radiographic teardrops should be symmetrical [9,10].

#### 2.3.3. Image Analyses

The images were arranged in random order irrespective of groups, and image analyses were conducted using a picture archiving and communication system workstation at our institution. A board-certified musculoskeletal radiologist, a board-certified interventional radiologist, and a fourth-year radiology resident participated as readers in the image review. A training session was conducted among the readers using images obtained from 50 patients not included in this study. The readers were provided with the image set and blinded to the original report, clinical data, and findings of other readers.

While reviewing the image set, each reader evaluated and recorded the status of acetabular sparing, the texture of the cup–bone interface, and the latitude cut deviation for each item. Acetabular sparing was considered “present” if the joint space and subchondral bone plate area were preserved [2,8]. The texture of the cup–bone interface was categorized as either smooth or textured [8]. The latitude cut deviation was graded in dichotomy based on the results of the first part of the investigation. Specifically, if the latitude cut was made around the equator of the outer circle (i.e., a lower percentage of latitude cut deviation), it was deemed to be “near the equator.” If the latitude cut was made markedly below the equator circle (i.e., with a higher percentage of latitude cut deviation), it was considered to have “deviated from the equator” (Figure 4).

#### 2.3.4. Statistics

Between-group differences in mean age and time interval between the date of surgery and radiography were determined using the Mann–Whitney U test. Sex, laterality, and imaging characteristics were analyzed using Fisher’s exact test. Receiver operating characteristic analysis was performed, and the resultant areas under the curve were used to compare the diagnostic performance of the imaging characteristics with respect to the identification of arthroplasty types (THA vs. hemiarthroplasty). Sensitivity, specificity, positive predictive value, negative predictive value, and accuracy were calculated. The inter-observer agreement among the three readers was evaluated using the Fleiss kappa coefficient (k), with k values of <0.20, 0.21–0.40, 0.41–0.60, 0.61–0.80, and 0.81–1.00 being interpreted as poor, fair, moderate, good, and excellent, respectively [11]. Statistical analyses were performed using SPSS version 25.0 (IBM Corp., Armonk, NY, USA) and R software version 4.2.0 (R Foundation for Statistical Computing, Vienna, Austria), with statistical significance set at *p* < 0.05.

## 3. Results

The image set included 20 hemiarthroplasty patients (9 females and 11 males; mean age: 71.3 years; range: 47–85 years) and 20 THA patients (6 females and 14 males; mean age: 59.7 years; range: 36–76 years). The mean time interval from the date of surgery to radiography was 60.5 months (range: 2–164 months) for hemiarthroplasty patients and 49.0 months (range: 8–138 months) for THA patients. The mean patient age significantly differed, reflecting a difference in indications for the respective operations. No statistically significant differences in sex or imaging time after surgery were observed (*p* > 0.05, Table 2). 

For all readers, the identification of two discriminators (namely, the texture of the interface and the latitude cut deviation) was significantly different between the hemiarthroplasty and THA patients (*p* < 0.001). Furthermore, acetabular sparing was significantly different for readers 1 and 3 (Table 2). The sensitivity, specificity, and accuracy in distinguishing THA from hemiarthroplasty were 100%, 90–100%, and 95–100% for the latitude cut deviation, 60–85%, 55–95%, and 62.5–77.5% for acetabular sparing, and 100%, 50–80%, and 75–90% for the texture of the cup–bone interface, respectively (Table 3).

The receiver operating characteristic curves consistently indicated that the latitude cut deviation had the highest area under the curve (0.950–1.000), followed by the texture of the cup–bone interface (0.750–0.900) and acetabular sparing (0.625–0.775) (Table 4). The difference in diagnostic performance between the latitude cut deviation and acetabular sparing was significant for all readers (*p* < 0.001). The mean areas under the curve for the texture of the cup–bone interface and the percentage of latitude cut deviation were statistically different between readers 1 and 2 (*p* < 0.001 and *p* = 0.012, respectively). However, the difference in diagnostic performance between the texture of the cup–bone interface and acetabular sparing was significant only for reader 2 (*p* = 0.029). 

The inter-observer agreement among the three readers ranged from moderate to excellent. The Fleiss kappa values were 0.499 for acetabular sparing, 0.772 for the texture of the cup–bone interface, and 0.900 for the latitude cut deviation.

## 4. Discussion

This present study revealed that the latitude cut deviation had excellent diagnostic performance and inter-reader agreement in distinguishing hemiarthroplasty from THA on radiographs.

The difference in the latitude cut deviation arises from the design principle that leaves one interface of articulation unconstrained in primary hip arthroplasty unless there is a special purpose [12]. Theoretically, a complete link between the femur and pelvis provides a substantial advantage for preventing dislocation, which represents the third most common complication of THA and can immensely jeopardize patients’ quality of life when it repeatedly occurs [13,14,15]. Such a link largely transfers the multiaxial force generated by patients’ complex movements to the bone–prosthesis interface, thereby increasing the risk of loosening [13,14,15]. Additionally, the liner structure comprehensively surrounding the artificial femoral head leads to a reduction in the impingement-free range of motion [13,14,15]. This type of implant is referred to as “THA with a constrained liner” [13,14,15] (Figure 1 and Figure 5). Because the general risk of dislocation is <1% in primary hip arthroplasty, the trade-off between the potential risk of dislocation versus the definitive increase in the risk of loosening and decrease in the impingement-free range of motion and loosening becomes unbalanced in general hip arthroplasty candidates [13,14,15]. Consequently, the constrained liner is selectively chosen for revision hip arthroplasty in patients with recurrent prosthetic dislocation. The primary choices for both hemiarthroplasty and THA maintain an articulating interface unconstrained, effectively preventing the transmission of multiaxial forces and reducing the impingement-free range of motion. In THA, the acetabular cup has a hemispherical geometry that allows the femoral head to move away from the liner surface in the axial direction (Figure 5) [16]. In hemiarthroplasty, the interface between the natural acetabular facet and the cup allows subluxation, whereas the artificial femoral head is captured in the metal-backed polyethylene cup–liner construct (Figure 5) [17,18]. Thus, the latitude cut of the spherical geometry in hemiarthroplasty is made below the equator to ensure that the cup–liner construct captures the inner head ball. In hemiarthroplasty, creating a cup–liner construct in a hemispherical geometry that does not capture the inner head ball leaves two subluxable interfaces and renders the cup–liner construct unstable (Figure 5). The difference in cup geometry based on the type of hip arthroplasty originates from the design rationale of arthroplasty and is also reflected on radiographs. The current study attempted to highlight this difference in images using the latitude cut deviation.

A previous report suggested that the prosthesis head in hemiarthroplasty was slightly larger than the acetabular cup in THA [8]. However, a specific method for comparison and a cut-off for discriminating between hemiarthroplasty and THA on radiographs were not provided.

In this present study, discriminating between the two types of hip arthroplasty based on joint space narrowing or widening around the cup and/or preservation of the acetabular subchondral plate yielded the lowest diagnostic performance and inter-reader reliability among the three tested criteria. A preserved subchondral bone plate of the acetabulum and an adjacent joint space can be utilized to distinguish hemiarthroplasty from THA [2,8,19]. Under normal circumstances, this serves as a reliable indicator for distinguishing hemiarthroplasty from THA, as the fundamental difference lies in whether the acetabular articular surface is replaced. Nevertheless, when the acetabular cartilage is worn down to the extent that the outer cup of the hemiarthroplasty contacts or protrudes into the subchondral bone, resulting in acetabular erosion and migration, the absence of the adjacent cartilage joint space could lead readers to mistakenly interpret such cases as an acetabular cup placed in a reamed acetabulum in THA [7,8,19] (Figure 6a). Furthermore, in THA cases, instances of aseptic loosening or osteolysis at the bone–prosthesis interface may be misinterpreted as the adjacent cartilage joint space observed in hemiarthroplasty [2] (Figure 6b).

The texture of the head/cup can be utilized to identify the distinctive features of hip arthroplasty, such as the smooth outer surface of the head/cup in hemiarthroplasty as compared with the textured one in THA [8]. Some acetabular components of THA possess a rough macrostructure on the surface of the cup, rendering the texture of the head/cup on radiographs a valuable parameter for determining the type of hip arthroplasty. However, certain acetabular components of THA incorporate only the microstructure for osteointegration without a rough macrostructure. In such cases, the texture of the cup–bone interface in THA often exhibits a smooth contour on radiographs, which contributed to the reduced specificity of this criterion in this study.

Definite signs such as a visible screw or screw hole on the acetabular cup or a prominently rough texture of the cup–bone interface were applied as exclusion criteria for the present study. The presence of a screw to enhance acetabular cup fixation can serve as a definitive distinguishing factor. Nonetheless, because acetabular screws are not universally present in THA, relying solely on their presence can lead to numerous instances in which THA lacking acetabular screws is inaccurately identified as hemiarthroplasty. 

In contrast to the conventional signs used to differentiate arthroplasty types, which could also indicate a complication of the opposite type, intrinsic geometrical differences offer a clear criterion for distinguishing hemiarthroplasty from THA. Consequently, this criterion enables the identification of specific complications associated with each type of arthroplasty.

In our study, two cases of THA were erroneously classified as hemiarthroplasty based on the criterion of latitude cut deviation (Figure 7). The boundary of the artificial femoral head could be inaccurately perceived as the cup margin. Historically, femoral heads ranging from 22 mm to 28 mm in diameter have been predominantly used for THA. However, recent advancements in bearing materials have enabled the use of thinner liner designs, thereby allowing the accommodation of larger artificial femoral heads. Along with the theoretical advantages associated with larger femoral heads, this has spurred the adoption of femoral heads with diameters ranging from 36 mm to 40 mm for THA. When a sizable femoral head aligns with the anteverted acetabular cup contour, it can bewilder readers lacking familiarity with hip implants. This perplexity may lead to a misinterpretation of the cup margin and implant outline. To mitigate this confusion, adhering to the principle of drawing a chord line that links the two points where the silhouette deviates from the outer sphere contour could facilitate a precise estimation of the latitude cut.

Our study has some limitations. First, this was a retrospective study that included a small number of patients who were selected through a consensus between a radiologist and an orthopedic surgeon. Our study aimed to compare the diagnostic performance of imaging characteristics in distinguishing hemiarthroplasty from THA. Given the low incidence of acetabular erosion after hemiarthroplasty, it was necessary to select a specific patient group to classify it with the same weightage as THA with/without acetabular erosions to ensure comparability of cases. Nevertheless, to the best of our knowledge, this is the first study to explore the inherent geometrical differences in prostheses that could be used to distinguish hemiarthroplasty from THA. Further studies involving a larger number of patients and various types of prostheses should examine the utility of the latitude cut deviation. Second, considering the wide array of types used worldwide, the inclusion of every existing hip implant was not feasible. However, we endeavored to include as many prostheses as possible in the first part of our investigation and incorporated all cups from the top three manufacturers into our investigation. Finally, evaluation of the latitude cut deviation alone cannot replace surgical records or sequential radiographs. If available, scrutinizing surgical records or comparing serial images is imperative to accurately assess the type of hip arthroplasty and postoperative complications. Our proposed method for “latitude cut deviation” provides radiologists and orthopedic surgeons with a concise means of identifying the type of arthroplasty.

## 5. Conclusions

The latitude cut deviation exhibited excellent diagnostic performance and inter-reader agreement in distinguishing hemiarthroplasty from THA on radiographs. 

To distinguish between different types of hip arthroplasty, an inherent feature within the implant contour was derived via an investigation of the prosthesis’s design rationale. The derived method for interpreting radiographs, named the “latitude cut deviation”, reduces the need for additional data beyond single-point images, such as surgical records or previously captured serial radiographs. By simplifying the determination of prosthesis type on radiographs, it facilitates clinicians in promptly recognizing implant-specific complications.

Further studies involving a larger number of patients and various prosthesis types should be conducted to ascertain whether this method can be widely employed to distinguish hemiarthroplasty from THA. Building upon the current methodology via an exploration of implant design rationale and its scientific validation, it is anticipated that in the future, similar approaches to image interpretation will be further proposed and discovered.

## Figures and Tables

**Figure 1 jcm-12-06218-f001:**
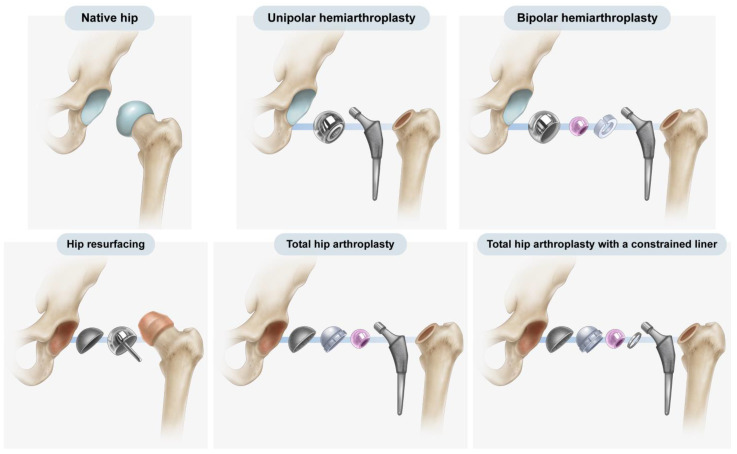
Explosion views of various types of hip arthroplasty. Upper left: The native hip joint possesses an articular surface on both the acetabulum and femoral head. Upper middle and right: Hemiarthroplasty entails the replacement of the original femoral head and neck with an artificial implant while leaving the native acetabular cartilage untouched. The distinction between unipolar and bipolar hemiarthroplasty comes down to the presence of an additional articulation inside the spherical component. Lower right: Conversely, THA involves the substitution of both the femoral and acetabular surfaces with artificial prostheses. Note that the acetabular cartilage remains intact in hemiarthroplasty but is removed to induce cup fixation in THA. Lower right: In THA, a constrained liner features a locking ring that secures the artificial femoral head in place while still permitting rotation. Lower left: In hip resurfacing, only the articular surface on both the acetabulum and femoral head is replaced.

**Figure 2 jcm-12-06218-f002:**
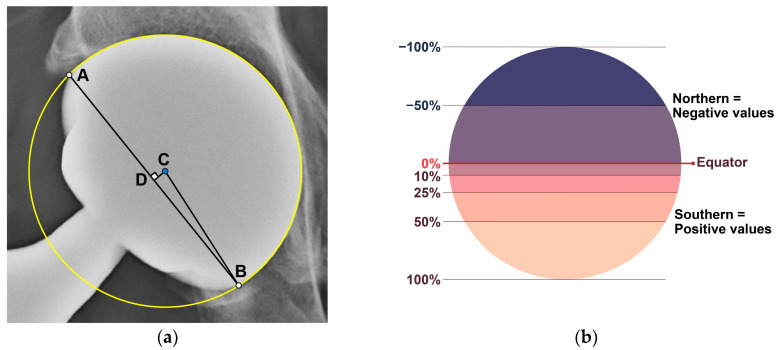
Diagram illustrating the quantitative measurement method for the latitude cut deviation (**a**) and examples of the percentage of latitude cut based on the relative position of the chord line within the circle (**b**). (**a**) After drawing a circle that best matches the outer contour of the sphere, two points are determined where the silhouette leaves the circle (point A & B). Point C represents the center of the circle, while point D is a point on the chord line that is closest to point C. The latitude cut deviation is defined as the relative position of the chord line to the sphere’s equator. This is quantitatively measured in terms of the percentage of latitude cut deviation, represented as CD/CB, which is equal to sin (∠ABC). (**b**) The degree of deviation of the chord line from the equator is visualized for each quantitatively expressed “percentage of latitude cut deviation”.

**Figure 3 jcm-12-06218-f003:**
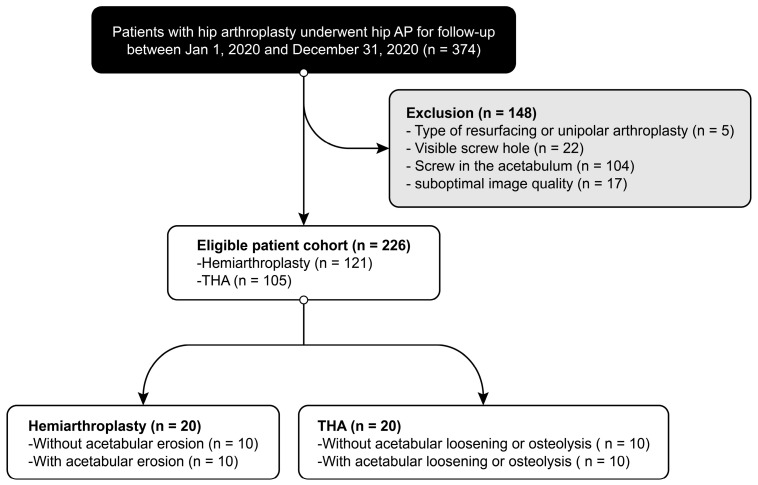
Flowchart illustrating patient inclusion and exclusion. AP, anteroposterior; BPHA, bipolar hemiarthroplasty; THA, total hip arthroplasty.

**Figure 4 jcm-12-06218-f004:**
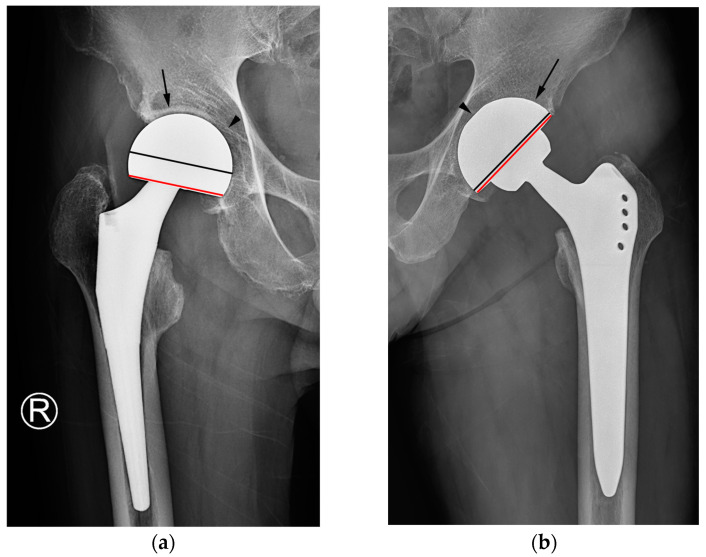
Simple radiographs of hemiarthroplasty (**a**) and THA (**b**). In bipolar hemiarthroplasty (**a**), the subchondral bone plate of the acetabulum and the adjacent joint space (arrow) are preserved, and the outer surface of the cup appears smooth instead of rough (arrowhead). The chord line of the cup (red line) deviates from the sphere’s equator. In THA (**b**), the subchondral bone endplate and cartilage space are absent (arrow), and the interface between the cup and bone is smooth, displaying a subtle indentation of the outer spherical contour (arrowhead). The chord line of the cup (red line) closely approaches the equator of the sphere (black line). In both images, the black line represents the equator of the sphere, whereas the red line indicates a chord line connecting the medial and lateral edges of the acetabular cup or bipolar head.

**Figure 5 jcm-12-06218-f005:**
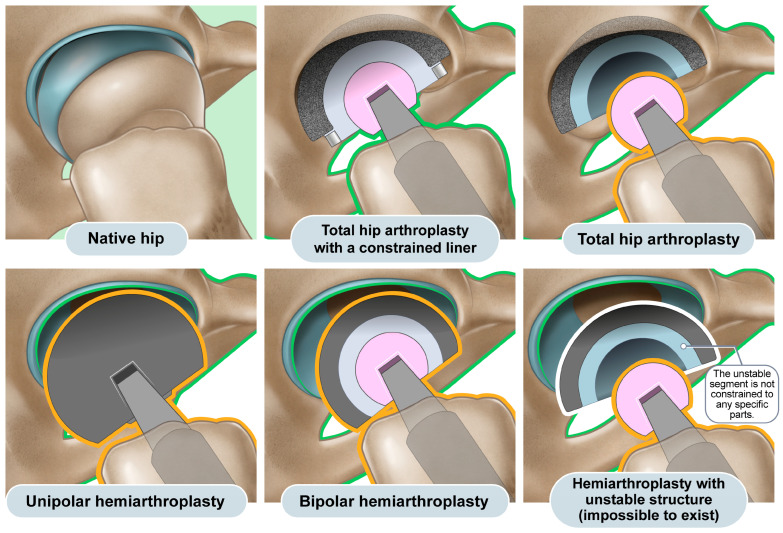
Cross-sections illustrating the relationship between components in various types of hip arthroplasty. The outline color represents the boundaries of components mechanically constrained within the construct. Green indicates a mechanically linked construct to the pelvis, whereas orange indicates a mechanically linked construct to the femur. Upper left: The concave acetabulum articulates with the spherical femoral head to form the native hip joint. Upper middle: The constrained liner comprehensively encases the artificial femoral head, allowing only rotation and preventing displacement, resulting in an axially linked state. Upper right, lower left and middle: In contrast, both THA and hemiarthroplasty leave one articulation unconstrained, allowing for displacement. Lower right: If the cup of the hemiarthroplasty does not adequately encompass the artificial femoral head, resulting in two unconstrained articulating interfaces, the cup–liner construct remains an unstable segment without constraint to either the pelvis or femur.

**Figure 6 jcm-12-06218-f006:**
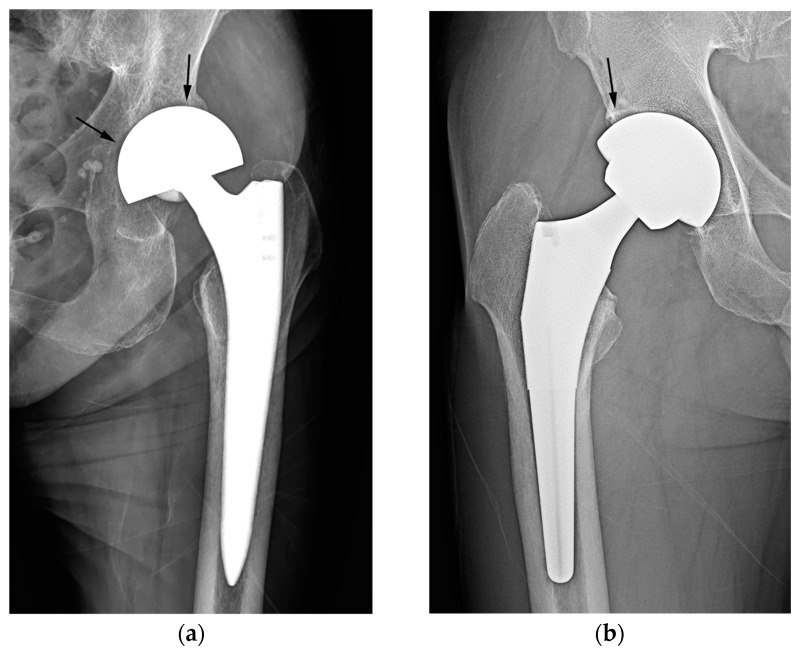
Misinterpreted cases of acetabular sparing. (**a**) A simple radiograph after hemiarthroplasty showing the absence of the subchondral bone endplate and cartilage space (acetabular erosion of the bipolar cup) (arrows). When the presence of the adjacent cartilage joint space was used as the discriminator, the case was misinterpreted as THA by all readers. (**b**) Simple radiography after THA showing the preserved joint space and a subchondral bone plate of the acetabulum (arrow). When the presence of the adjacent cartilage joint space was used as the discriminator, two readers misinterpreted the case as a hemiarthroplasty.

**Figure 7 jcm-12-06218-f007:**
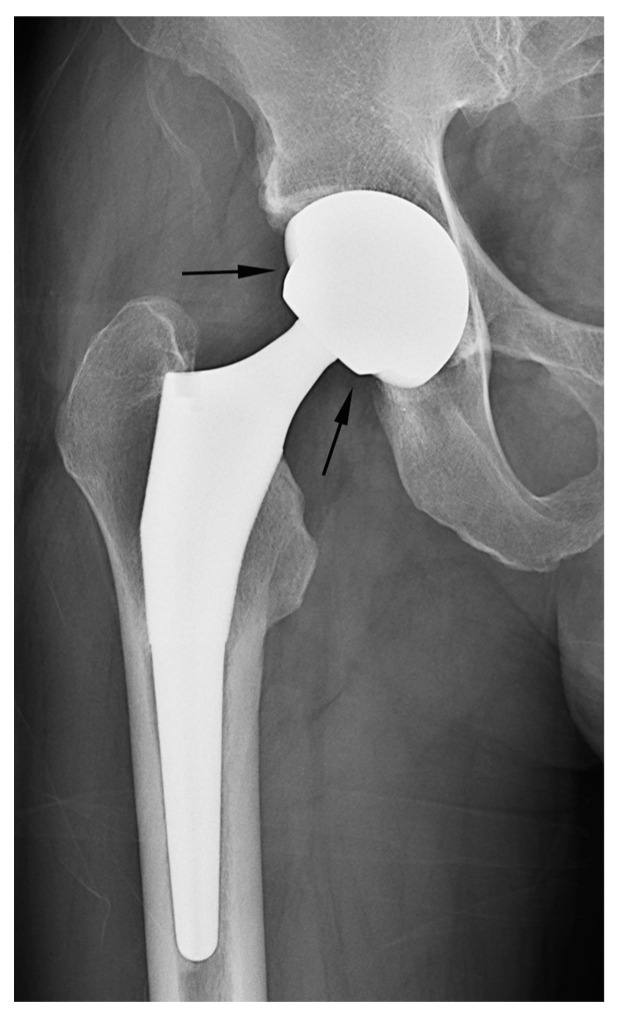
A simple radiograph after THA showing that the margin of the artificial femoral head (arrow) was confused with the margin of the cup by the raters. Adhering to the principle of determining a chord line that connects the points where the silhouette diverges from the outer sphere contour can facilitate an accurate estimation of the latitude cut.

**Table 1 jcm-12-06218-t001:** Percentage of latitude cut deviation in acetabular and bipolar cup designs according to cup size.

Type of Arthroplasty Cup	Manufacturer	Brand	Cup Size																																		
			36	38	39	40	41	42	43	44	45	46	47	48	49	50	51	52	53	54	55	56	57	58	59	60	61	62	64	66	68	70	72	74	76	78	80
Bipolar cup	DePuy Synthes	Self-Centering Hip			27%	26%	22%	20%	34%	32%	30%	34%	33%	31%	27%	24%	22%	22%	23%	21%	20%	19%															
	Corentec	Bencox Bipolar Cup		24%	23%	22%	22%	21%	23%	27%	27%	26%	26%	25%	25%	24%	24%	23%	23%	22%	22%	21%	21%	21%	20%	20%											
	Stryker *	UHR Universal Head		36%	31%	28%	30%	28%	30%	28%	28%	28%	27%	28%	28%	28%	28%	28%	28%	28%	28%	28%		28%			27%		27%		26%		25%				
	Zimmer Biomet *	Multipolar Cup		34%	33%	37%	34%	33%	33%	42%	40%	36%	39%	37%	36%	40%	33%	33%	33%	32%	32%	32%	31%	31%		30%		29%									
Total hip arthroplasty cup	DePuy Synthes	Pinnacle												0%		0%		0%		0%		0%		0%		0%		0%	0%	0%							
		DuraLoc												−13%		−12%		−12%		−11%		−11%		−11%		−10%		−10%	−10%	−9%							
	Corentec	Bencox										9%		3%		1%		5%		1%		1%		1%		1%		1%	1%	1%	1%						
		Hybrid										9%		3%		1%		5%		1%		1%		1%		1%		1%	1%	1%	1%	1%	1%	1%			
		Mirabo										6%		1%		1%		6%		1%		−1%		−1%		−1%		−1%	−1%	−1%	−1%	−1%	−1%	−1%			
	Zimmer Biomet *	Continuum								4%		5%		5%		6%		6%		5%		5%		5%		4%		4%	4%	4%	3%						
		Trilogy	4%	3%		2%		1%		1%		1%		1%		1%		1%		1%		1%		0%		0%		0%	0%	0%	0%	0%	0%	0%	0%	0%	0%
		G7						5%		3%		4%		4%		3%		3%		4%		4%		3%		3%		3%	3%	3%	3%						
	Stryker *	Trident				6%		9%		6%		6%		5%		5%		4%		3%		3%		3%		3%		3%	3%	3%	3%	3%	3%				
		Trident II						0%		0%		0%		0%		0%		0%		0%		0%		0%		0%		0%	0%	0%	0%	0%	0%				

* Digitally measured from scanned images of acetate templates.

**Table 2 jcm-12-06218-t002:** Clinical information and identification of three discriminators (acetabular sparing, texture of the interface, and latitude cut deviation) for determining the type of hip arthroplasty.

Variables of Interest	Hemiarthroplasty	THA	*p*-Value
Clinical information			
Age (years)	71.3 ± 11.2	59.7 ± 12.3	**0.003**
Sex			0.514
Male	11	14	
Female	9	6	
Laterality			1
Right	11	12	
Left	9	3	
Imaging time after arthroplasty (months)	60.5 ± 50.1	49 ± 39.1	0.424
Imaging characteristics			
Reader 1			
Acetabular sparing			**0.019**
Present	17	9	
Absent	3	11	
Texture of the interface			<**0.001**
Smooth	20	10	
Rough	0	10	
Latitude cut deviation			<**0.001**
Deviated from the equator	20	0	
Near the equator	0	20	
Reader 2			
Acetabular sparing			0.205
Present	12	7	
Absent	8	13	
Texture of the interface			<**0.001**
Smooth	20	7	
Rough	0	13	
Latitude cut deviation			<**0.001**
Deviated from the equator	20	2	
Near the equator	0	18	
Reader 3			
Acetabular sparing			<**0.001**
Present	12	1	
Absent	8	19	
Texture of the interface			<**0.001**
Smooth	20	4	
Rough	0	16	
Latitude cut deviation			<**0.001**
Deviated from the equator	20	1	
Near the equator	0	19	

Data are presented as means ± standard deviations for age and time interval; THA, total hip arthroplasty. Bold text indicates *p*-values below 0.05, signifying statistical significance.

**Table 3 jcm-12-06218-t003:** Sensitivity, specificity, and accuracy of the three indicators for identifying the type of arthroplasty based on the interpretations of three readers.

Readers	Sensitivity (%)	Specificity (%)	Accuracy (%)
Reader 1			
Acetabular sparing	17/20 (85)	11/20 (55)	28/40 (70)
Texture of the interface	20/20 (100)	10/20 (50)	30/40 (75)
Latitude cut deviation	20/20 (100)	20/20 (100)	40/40 (100)

Reader 2
Acetabular sparing	12/20 (60)	13/20 (65)	25/40 (62.5)
Texture of the interface	20/20 (100)	13/20 (65)	33/40 (82.5)
Latitude cut deviation	20/20 (100)	18/20 (90)	38/40 (95)

Reader 3
Acetabular sparing	12/20 (60)	19/20 (95)	31/40 (77.5)
Texture of the interface	20/20 (100)	16/20(80)	36/40 (90)
Latitude cut deviation	20/20 (100)	19/20 (95)	39/40 (97.5)

**Table 4 jcm-12-06218-t004:** Area under the ROC curve for the three discriminators based on the interpretations of three readers.

	Mean Area under the ROC Curve			95% Confidence Interval			*p*-Value		
	Acetabular Sparing (A)	Texture of the Interface (B)	Latitude Cut Deviation (C)				A vs. B	A vs. C	B vs. C
Reader 1	0.700	0.750	1.000	0.535–0.834	0.588–0.873	0.912–1.000	0.527	**<0.001**	**<0.001**
Reader 2	0.625	0.825	0.950	0.458–0.773	0.672–0.927	0.831–0.994	**0.029**	**<0.001**	**0.012**
Reader 3	0.775	0.900	0.975	0.615–0.892	0.763–0.972	0.868–0.999	0.111	**<0.001**	0.067

ROC, receiver operating characteristic. Bold text indicates *p*-values below 0.05, signifying statistical significance.

## Data Availability

The datasets used and/or analysed during the current study available from the corresponding author on reasonable request.

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
