# Peer review of "Deviation of Latitude Cut: A Simple Sign to Differentiate Total Hip Arthroplasty from Hemiarthroplasty in Radiography"

_jcm, 2023, doi:10.3390/jcm12196218_

Round 1

Reviewer 1 Report

• After using long versions with abbreviations, long versions should not be used again (For example: total hip arthroplasty (THA))

• The difference between the current study and the literature should be clearly mentioned in the Introduction section.

• Conclusions section should be expanded. The importance of the current study and what advantages it can provide to researchers for future studies should be mentioned.

• Explanations can be enriched by increasing references. (Especially 2023 and 2022)

• Small size texts in Figure 2b should be made visible.

• Figure 3 should be modernized.

Reviewer 2 Report

Title:   Deviation of latitude cut: A simple sign to differentiate total hip arthroplasty from hemiarthroplasty in radiography

Abstract: Okay.

Keywords:  Okay.

Introduction: Appropriate. This study proposes a latitude cut deviation for identifying hip arthroplasty types and to test its diagnostic value in radiography in separating total hip arthroplasty (THA) from hemiarthroplasty.

Material& methods: Radiologic examination was done with standard AP hip 
radiographsThree readers independently evaluated radiographs.

Discussion: Appropriate but since the sphericity and depth of the acetabular hemisphere are entirely different, a familiar eye can make a diagnosis without measuring.

Conclusion: Appropriate. The latitude cut deviation on radiographs shows good diagnostic performance and interreader agreement in identifying hemiarthroplasty from THA. It may be meaningful for the radiologists(!) to be able to make this discrimination clearly on the radiograph. However, orthopedic surgeons (we) usually have no trouble distinguishing HA and THA. Anteversion angle and inclination could be measured to understand acetabular component positioning 
Please correct the misspellings.
References: Up to date. 
The article has been evaluated in general terms.  It is technically acceptable but there is nothing to add literature for orthopedic surgeons. I appreciate the efforts of the authors.
Best Regards.
​
